# Irisin Is Related to Non-Alcoholic Fatty Liver Disease (NAFLD)

**DOI:** 10.3390/biomedicines10092253

**Published:** 2022-09-11

**Authors:** Marcin Kosmalski, Józef Drzewoski, Izabela Szymczak-Pajor, Andrzej Zieleniak, Melania Mikołajczyk-Solińska, Jacek Kasznicki, Agnieszka Śliwińska

**Affiliations:** 1Department of Clinical Pharmacology, Medical University of Lodz, 90-153 Lodz, Poland; 2Central Teaching Hospital of Medical University of Lodz, 92-213 Lodz, Poland; 3Department of Nucleic Acids Biochemistry, Medical University of Lodz, 92-213 Lodz, Poland; 4Department of Structural Biology, Medical University of Lodz, 92-213 Lodz, Poland; 5Department of Internal Medicine, Diabetology and Clinical Pharmacology, Medical University of Lodz, 92-213 Lodz, Poland

**Keywords:** irisin, non-alcoholic fatty liver disease, diagnosis, pathophysiology

## Abstract

Irisin is a cytokine involved in many metabolic pathways occurring, among others, in muscles, adipose tissue and liver. Thus, fluctuations in irisin levels are suggested to be related to metabolic diseases. Therefore, the purpose of our study was to evaluate whether irisin may be associated with non-alcoholic fatty liver disease (NAFLD). A total of 138 patients (70/68 male/female, mean age 65.61 ± 10.44 years) were enrolled in the study. The patients were assigned to the NAFLD group (*n* = 72, including 46 patients with type 2 diabetes (T2DM]) and the group without NAFLD (*n* = 66, 31 patients with T2DM). NAFLD was diagnosed based on ultrasound examination, Hepatic Steatosis Index (HSI) and Fatty Liver Index. Baseline anthropometric, blood pressure and biochemical parameters were collected. The serum irisin level was determined using an ELISA test. We observed that NAFLD was associated with an increased concentration of irisin. Moreover, Spearman correlations and linear regression analysis revealed that irisin level correlates with some anthropometric and biochemical parameters such as body mass index, glycated hemoglobin, aspartic aminotransferase, creatinine and urea. Logistic regression analysis depicted that odds for NAFLD increase 1.17 times for each 1 μg/mL rise of irisin concentration. Finally, ROC analysis showed that the concentration of irisin possesses a discriminate capacity for NAFLD and optimal cut points concentration could be designed. The risk of NAFLD in the subgroup with irisin concentration above 3.235 μg/mL was 4.57 times higher than in patients with the lower concentration of irisin. To conclude, the obtained results suggest that irisin concentration is associated with some anthropometric and biochemical parameters and should be further investigated toward its usage as a diagnostic biomarker of NAFLD.

## 1. Introduction

Non-alcoholic fatty liver disease (NAFLD) is one of the most common liver pathologies and is the main reason for patients reporting to hepatologists [1]. The disease is diagnosed when macrovesicular steatosis is found in ≥5% of hepatocytes, excluding other secondary causes of fatty liver infiltration [2]. NAFLD is considered a metabolic disorder that results from complex interactions between nutrients, hormones and genetic factors. A Western lifestyle, a high calorie-rich diet, and decreased physical activity are among the most crucial factors in the development of NAFLD [3]. It has been suggested that insulin resistance (IR) accompanied by hyperinsulinemia, which leads to an increase in lipolysis in adipocytes and an elevated level of circulating free fatty acids captured by hepatocytes, plays a pivotal role in the development of NAFLD. These abnormalities trigger the initiation of disturbances in many metabolic pathways, including fats (de novo hepatic lipogenesis), carbohydrates (glycogenolysis and gluconeogenesis), and changes in the profile of cytokines and adipokines released by adipose tissue, leading to the development of chronic inflammation [4]. 

The presence of NAFLD is associated with the risk of health consequences. They result from both the natural progression of liver disease (non-alcoholic steatohepatitis—NASH, cirrhosis and hepatocellular carcinoma—HCC) and an increased risk of developing other pathologies, such as cardiovascular complications (cardiovascular disease—CVD, including heart failure, atrial fibrillation and chronic subclinical and clinical coronary syndrome) and non-HCC malignant neoplasms (including mainly the large intestine, breast, prostate, lung and pancreas) [5,6]. Recently, attention has also been paid to the relationship between NAFLD and NAFLD-related chronic kidney disease (CKD) and the impact of NAFLD on its severity, regardless of established cardiovascular risk factors [7]. It should also be noted that NAFLD is closely related to the presence of carbohydrate disorders, including type 2 diabetes (T2DM), both in epidemiological and etiological terms [8]. It has been shown that the coexistence of these two pathologies in one patient increases the risk of hepatic complications (non-alcoholic steatohepatitis—NASH, NAFLD-related hepatocellular cancer—NAFLD-HCC, liver cirrhosis), diabetic complications (both micro- and macrovascular) and CVD [9,10,11].

NAFLD diagnostics raise much controversy. To date, liver biopsy remains the gold standard procedure for the diagnosis of NASH and the staging of NAFLD. The usefulness of this diagnostic tool in daily medical practice raises much controversy, especially in view of the fairly good prognosis for most NAFLD patients, the lack of a definitively established form of therapy and the risks and costs associated with performing this test. Regardless of the non-invasive NAFLD diagnostic tool used, other secondary causes of fatty infiltration of the liver should be ruled out first, including alcohol abuse, certain medications, genetic and chronic diseases, and environmental factors [12]. To assess fat infiltration, both imaging tests (ultrasound—US, transient elastography, computed tomography, magnetic resonance—MR and Xenon-133 liver scan) and laboratory tests to calculate specific scales (for example NAFLD liver fat score—NLFS, lipid accumulation product—LAP, SteatoTest, NAFL screening score and Fatty Liver Index—FLI or Hepatic Steatosis Index—HSI) are used. Each of these diagnostic methods is characterized by specific sensitivity and specificity in the diagnosis of NAFLD [13,14]. 

Irisin is a hormone involved in metabolic pathways; namely, it increases the uptake of glucose and fatty acids by muscles, reduces gluconeogenesis and stimulates glycogenesis in the liver, as well as converts white adipose tissue (WAT) into brown adipose tissue. Irisin was also demonstrated to diminish the severity of inflammation as well as to affect the function of kidneys, neurons, bones, endothelial cells and beta cells of the pan-creas [15,16]. Many research groups have identified the association between irisin level and heart disease (heart failure, cardiac hypertrophy), hypertension, stroke, pulmonary diseases (chronic pulmonary disease, neonatal respiratory distress syndrome), CKD, Alzheimer disease, osteoporotic fractures and some cancers (prostate, hepatocellular, pancreatic). It has been suggested that irisin has a protective effect against these pathologies [17]. Irisin has been linked to favorable effects on metabolic disease, including obesity, T2DM, dyslipidemia and NAFLD [18]. The relationship between NAFLD and irisin is controversial, and recently published studies have indicated an increased concentration of this myokine in patients with fatty liver [19]. These differences may be due to the ELISA kits for the determination of irisin, patients included in the study and NAFLD diagnosis methods [20].

Therefore, the aim of this study was to determine whether irisin may be related to NAFLD. To achieve this goal, we assessed the correlation between the irisin levels determined by ELISA test and the prespecified anthropometric and metabolic parameters in groups of patients with and without NAFLD diagnosis due to ultrasound examination, HSI and FLI.

## 2. Materials and Methods

### 2.1. Characteristics of the Patients

In total, 138 adult patients aged 65.61 ± 10.44 years on average (including 68 women and 77 patients with T2DM) hospitalized between January 2016 and April 2016 for various internal medicine diseases at the Department of Internal Medicine, Diabetology and Clinical Pharmacology at the Medical University of Lodz were included in the study. The study was approved by the Bioethics Committee of the Medical University of Lodz. Prior to enrollment, each participant provided written consent to participate in the study. Exclusion criteria included the presence of secondary causes of excessive fatty liver infiltration (ethanol consumption more than 14 g/day; liver diseases; medications—chemotherapy, combined antiretroviral therapy, amiodarone, methotrexate, tamoxifen, corticosteroids, tetracyclines, valproic acids, amphetamines, acetylsalicylic acid, genetic causes, such as hemochromatosis, alpha-1 antitrypsin deficiency, Wilson’s disease, congenital lipodystrophy, abetalipoproteinaemia, hypobetalipoproteinaemia, familial hyperlipidaemia, lysosomal acid lipase deficiency, glycogen storage diseases, hereditary fructose intolerance, urea cycle disorders, citrin deficiency; environmental causes—lead, arsenic, mercury, cadmium, herbicides, pesticides, polychlorinated biphenyls, chloroalkenes; nutritional/gastroenterological causes—severe surgical weight loss, starvation, malnutrition, total parenteral nutrition, microbiome changes, celiac disease, pancreatectomy, short bowel syndrome and other causes such as hypothyroidism, polycystic ovary syndrome, hypothalamic or pituitary dysfunction, growth hormone deficiency, HELLP (hemolysis, elevated liver enzymes and low platelets), Amanita phalloides mushrooms, phosphorous poisoning, petrochemicals and Bacillus cereus toxin), prediabetes and type 1 diabetes mellitus, gestational or other than T2DM, diseases that may affect muscle metabolism such as glycogen metabolism disorders, lipid metabolism disorders and mitochondrial myopathies, severe kidney or liver dysfunction (including those who had had an organ transplant), musculoskeletal damage, or a surgery undergone during the last 6 months, pregnancy and severe infections. The enrolled patients had a medical history and underwent a physical examination. The necessary diagnostic tests for glucose metabolism abnormalities (according to ADA) or the presence of secondary causes of fatty livers were performed. Blood pressure (BP, including systolic blood pressure—SBP and diastolic blood pressure—DBP) and anthropometric measurements (body weight, height, waist (WC) and hip circumference (HC)) were measured and used to calculate the body mass index (BMI) and waist-hip ratio (WHR). Next, blood samples were taken to determine fasting plasma glucose (FPG), glycated hemoglobin level (HbA1c), total cholesterol (T-CH), LDL cholesterol (LDL-CH), HDL cholesterol (HDL-CH), triglycerides (TG), total bilirubin, uric acid, urea, creatinine concentrations, liver enzymes, including alanine aminotransferase (ALT), asparagine (AST) and gamma-glutamyltransferase (GGTP) activity. Postprandial plasma glucose (PPG) was measured two hours after standardized breakfast (equivalent to approximately ~20% of their total energy requirement). These markers were measured using standard laboratory methods. The estimated glomerular filtration rate (eGFR) was calculated based on the Modification of Diet in Renal Disease (MDRD) equations. An ultrasound examination (US) was also performed by an experienced radiologist to assess the presence of hepatic steatosis. Hepatic steatosis was assessed based on observations including (1) increased liver echogenicity compared to the renal cortex; (2) decreased conspicuity of hepatic vasculature; (3) presence of focal fat sparing; and (4) decreased ability to visualize the diaphragm and deeper liver parenchyma. We also calculated the HSI and FLI to confirm the presence of fatty liver by ultrasound examination. 

The HSI was calculated using the following index equation:

8 × ALT/AST + BMI (+2 if T2DM yes, +2 if female)

here BMI is calculated using: body weight (kg)/height squared (m^2^). HSI values of 36 and above indicate that the presence of NAFLD is highly likely. Values below 30 indicate that NAFLD can be ruled out [21]. 

The FLI was calculated from the formula:

(e0.953*loge (TG) + 0.139*BMI + 0.718*loge (GGTP) + 0.053*WC − 15.745)/(1 + e0.953*loge (TG) + 0.139*BMI + 0.718*loge (GGTP) + 0.053*WC − 15.745) × 100
(1)


FLI scores of 60 and above indicate that NAFLD is present, between 30 and 60 remain inconclusive and below 30 indicate that NAFLD should be ruled out [22].

The patients were assigned to the following groups: patients with concomitant NAFLD (+NAFLD; *n* = 72, including 46 patients with T2DM) and patients without NAFLD (−NAFLD; *n* = 66, including 31 patients with T2DM). 

If patients did not show evidence of fatty liver in all of these studies (US, HSI and FLI), the diagnosis of NAFLD seemed questionable and the patient was not enrolled in the study.

### 2.2. Determination of Irisin Concentration—ELISA Assay

After blood sampling, the serum was separated and frozen at a temperature of −80 °C. Serum concentration of irisin was determined using an ELISA Kit (BioVendor—Laboratorní medicína a.s. Brno, Czech Republic) according to the manufacturer’s protocol. The inter-assay variability and intra-assay variability were <7% and <8%, respectively.

### 2.3. Statistical Analysis

The anthropometric and biochemical characteristics as well as irisin concentration of the groups were expressed as medians with lower and upper quartiles. As the distribution of variables was not in accordance with normality determined by the Shapiro–Wilk test, the differences between 2 groups were assessed by the Mann–Whitney U test. A Chi square test was performed for sex ratio and T2DM ratio comparisons in the groups. The relationship between irisin concentration and anthropometric and biochemical parameters was evaluated using a Spearman (non-parametric) correlation coefficient. The most significant irisin-related variables were determined using the stepwise forward multiple regression method. Logistic regression analysis was conducted to assess the relationship between the risk of NAFLD and irisin concentration. The non-parametric variables included in the linear regression analysis were logarithmically transformed. Receiver Operating Characteristic (ROC) analysis with an area under the curve (AUC) as a quality index was carried out to determine the potential of irisin concentration as a diagnostic tool. The optimal cut-point was determined by means of the Younden index method (cut-point 3.235 μg/mL). The patients were divided into two subgroups according to the cut point of irisin concentration. The difference in frequency of NAFLD between the subgroups was tested by Chi square test with Yates correction. All analyses were performed using GraphPad Prism 8.0 software (San Diego, CA, USA) and STATISTICA 13.0 software (Statsoft, Tulsa, OK, USA). *p* < 0.05 was considered statistically significant.

## 3. Results

The comparison of the studied groups (+NAFLD and −NAFLD) in terms of age, sex, T2DM occurrence, anthropometric parameters and blood pressure is presented in Table 1. Table 2 displays the differences in biochemical parameters between the NAFLD and −NAFLD groups of patients. We found that patients with NAFLD did not differ in terms of age, sex, height, blood pressure values and the assessed markers of carbohydrate metabolism (FPG, PPG, HbA1c) or kidney function (creatinine, urea, eGFR), T-CH, LDL-CH, HDL-CH, total bilirubin, and uric acid concentrations from the −NAFLD group. As expected, patients with NAFLD had statistically significant higher anthropometric parameters (body weight, WC and HC, BMI, WHR), activities of liver injury markers (ALT, AST, GGTP), TG concentrations, and HSI and FLI values. Interestingly, the number of patients suffering from T2DM was statistically significantly greater in the +NAFLD group than in the −NAFLD group.

As depicted in Figure 1, we found a significantly higher level of irisin in the NAFLD group in comparison to the −NAFLD group (4.53 ± 2.62 μg/mL vs. 5.89 ± 3.53 μg/mL, *p* < 0.01).

Table 3 shows the results of the linear regression and correlation analysis between the irisin level and anthropometric and biochemical parameters in the whole study population. Irisin concentration positively correlated with BMI, HbA1c, eGFR and HSI and negatively with total bilirubin, creatinine, urea and uric acid. The results of multivariate stepwise linear regression analyses showed that BMI, creatinine and Hba1c values were most strongly and significantly corelated with irisin concentration.

Table 4 presents the associations between irisin concentration and the presence of NAFLD (with interactions). We observed increased irisin levels in patients with coexisting NAFLD and increased AST and diminished creatinine levels, as depicted in Model 1. In turn, Model 2 revealed increased irisin levels in patients with elevated urea, HbA1c and BMI. Finally, Model 3 shows increased irisin levels in patients with increased BMI and HbA1c, and decreased creatinine. These results at least partially further confirm the data presented in Table 3.

Table 5 shows the effect of irisin on the risk of NAFLD, as assessed by logistic regression analysis. The odds for NAFLD increases 1.17 times for each 1 μg/mL rise of irisin concentration (Model 1; unadjusted). In turn, model 2 adjusted with the most important irisin corelated biomedical parameters (BMI, creatinine and HbA1c), revealed that the risk of NAFLD was the most dependent on BMI. Irisin levels demonstrated a trend for the risk of NAFLD, but the relationship did not reach statistical significance. The effect of irisin with adjustment to BMI should exceed the threshold of statistical significance with an increasing number of patients in the study. HbA1c and creatinine did not exert any significant impact on NAFLD.

To explore the potential of irisin concentration as a diagnostic tool for NAFLD risk, ROC analysis was conducted (AUC = 0.643, *p* = 0.0023). The optimal cut-point was determined by means of the Younden index method (cut-point 3.235 μg/mL). The patients were divided into two subgroups according to the cut-point of irisin concentration. The difference in frequency of NAFLD between the subgroups was tested by the Chi square test with Yates correction (*p* = 0.0009). The risk of NAFLD in the subgroup with a higher irisin concentration was found to be above 4 times higher than that in the subgroup with a low concentration of irisin (odds: 1.52 and 0.33 respectively; OR = 4.57).

## 4. Discussion

The incidence and prevalence of NAFLD have increased worldwide. Recently, NAFLD has been reported to afflict more than 25% of the general population worldwide. The prevalence of this pathology is still varied, and differences result from region, race/ethnicity and diagnostic tool as well as age, sex and the presence of metabolic syndrome (obesity, dyslipidemia and T2DM) [23]. However, there are many methods for diagnosing NAFLD. The studies performed mainly used one or two tools to diagnose this pathology (biopsy, imaging biomarkers or appropriate scores basic due to age, sex, anthropometric parameters, presence of T2DM and some laboratory parameters) [24]. Recent studies indicate a high correlation between the US examination and the proposed cut-off points for the HSI and FLI scales (>36 and >60, respectively) [25,26]. Therefore, in our study, NAFLD was diagnosed based on the US study and the results of the HSI and FLI. Nevertheless, new NAFLD diagnosis markers are still being south, which are characterized by higher diagnostic sensitivity and specificity. This is linked with the necessity of identifying the causes leading to the excessive accumulation of lipids in the liver. An increasing number of studies have indicated an etiological relationship between NAFLD and irisin concentration [27].

Irisin is a peptide produced from proteolytic cleavage, glycosylation and probably dimerization of processing of Fibronectin type III domain-containing protein 5 (FNDC5) [28]. The expression of FNDC5 is under the control of the peroxisome proliferator activated receptor g coactivator 1-a, which plays a critical role in the maintenance of glucose, lipid and energy homeostasis [29,30,31]. Studies have shown that FNDC5 transcript is expressed in multiple tissues including the heart, pericardium, intracranial artery, lung, brain, optic nerve, spinal cord, retina, uvula, ovary, oviduct, seminal vesicles, testis, vagina, urethra, urinary bladder, penis, adrenal gland, pituitary, kidney, esophagus, vena cava, stomach, tongue, rectum, small intestine, tonsil, thyroid and liver [32]. The most significant roles of irisin include browning white adipocytes, neural proliferation, and bone metabolism. Other effects of irisin involve glucose homeostasis and cardiac metabolism, which are still under investigation [33].

We found that NAFLD patients had elevated irisin levels. To determine the potential of irisin concentration as a diagnostic tool for NAFLD risk, we conducted multiple statistical analyses, including logistic regression and ROC analysis. Our results have shown that the risk of NAFLD is above 4 times higher if irisin concentrations are higher than 3.235 μg/mL. Additionally, we found that NAFLD increases 1.17 times for each 1 μg/mL rise of irisin concentration. Data on irisin concentrations in NAFLD patients are inconsistent and leave a lot of confusion. Most studies indicate a decreased concentration in this group of patients [34,35,36]. However, some studies have shown elevated irisin levels in NAFLD patients [19,37]. There are also data showing no effect on NAFLD on the concentration of this myokine [38]. It has been suggested that these discrepancies may be due to methods of diagnosing fatty liver, the analyzed group of patients and methods of determining irisin.

In studies assessing the concentration of irisin, imaging tests (US, MR) or liver biopsies are the main methods for diagnosing fatty liver infiltration [19,34,35,36,37,38,39]. In our study, we made the diagnosis of NAFLD based not only on US but also on HSI and FLI. There are no studies assessing the concentration of irisin in patients diagnosed with NAFLD based on simple non-invasive and non-image scales. These scales are widely used in everyday outpatient medical practice due to their minimally invasive nature, safety, general availability and easy implementation on a large scale. The available data shows differences in the detection scores of NAFLD. Lind et al. found that FLI may be preferable in a population-based sample [40]. The recently suggested diagnostic and prognostic values of HSI and FLI should also be mentioned [25,41]. In our study, we expanded the diagnostics to include the US test, which is also a cheap, non-invasive and easily available test to enhance the diagnostic value of NAFLD.

FNDC5 and irisin have been detected mainly by 3 antibody-dependent methods: Western blot (qualitative and semi-quantitative), enzyme-linked immunosorbent assays (ELISA, quantitative) and protein liquid chip assay (Milliplex Map Human Myokine Magnetic Bead Panel; Merck, Darmstadt, Germany, quantitative). Among all the methods, mass spectrometry (qualitative and quantitative) with quantification peptides is considered the “gold standard” for the measurement of protein concentrations. However, the credibility of this method raises a lot of controversy because high methodological variability, preparation of plasma or serum samples for MS requires removal of highly abundant albumins and immunoglobulins and subsequent concentration, which leads to varying amounts of retained proteins for analysis and the very low MS values question the validity of a large proportion of all reported irisin levels measured with ELISA [42]. In our study, the ELISA Kit (BioVendor—Laboratorní medicína a.s. Czech Republic) was used to determine irisin concentration, as in other studies of NAFLD patients [19,34,35,36,37,38,39]. It should be emphasized that, in NAFLD patients, no other methods of administering irisin than ELISA were used, which have been used in patients with other than hepatic steatosis diseases [43,44].

The study population, which differs in age, sex, obesity or overweight, presence of dyslipidemia, T2DM/prediabetes, physical activity or other comorbidities, may have a significant impact on irisin concentration. It should be noted that when other causes of fatty liver were excluded, different amounts of alcohol were considered [34,35,37,39] or the amounts were not specified [19,36,38].

In our study, patients with NAFLD had statistically higher body weight, WC, HC, BMI, WHR, ALT, AST, GGTP and TG than patients without NAFLD. Interestingly, the number of patients suffering from T2DM was statistically significantly greater in the +NAFLD group than in the −NAFLD group. There were no statistically significant differences between the groups in age, sex, height, blood pressure, markers of carbohydrate metabolism (FPG, PPG, HbA1c), kidney function (creatinine, eGFR, urea), total bilirubin, T-CH, LDL-CH, HDL-CH and uric acids. In Polyzos’s study, NAFLD patients did not differ in age, BMI, WC, HDL-CH, uric acids and glucose concentration, but had higher AST, ALT, GGTP and TG values than patients without NAFLD. In linear regression analysis, irisin was independently inversely associated after adjustment for BMI or WC, gender and age in the NAFLD group [38]. Another study by the same author found significantly higher BMI, WC, ALT, AST, GGTP and TG values in the NAFLD group [34]. On the other hand, Shanaki et al. found differences in values of BMI, WHR, DBP, AST, ALT, GGTP and creatinine, but not in age, SBP, FPG, T-CH, TG, HDL-CH, LDL-CH and urea concentrations between +NAFLD and −NAFLD patients. The parameters inversely associated with irisin levels included BMI, WHR, FPG and liver enzymes (ALT, AST, GGTP) [45]. Choi et al. confirmed that the presence of fatty liver is associated with higher BMI, WC, SBP, SDP, FPG TG, HDL-CH, LDL-CH and AST, ALT. It should be mentioned that in this study, a comparable group of patients (with and without NAFLD) differed significantly in terms of size and sex. Interestingly, they found that NAFLD patients presented inverse correlations between irisin concentration and fat content in the liver (higher in the mild fatty liver group than in the moderate-to-severe fatty liver group). Additionally, they did not find a correlation with the quantity of exercise, TG, AST and ALT levels [37]. Ulualan et al. conducted their study on a group of children aged 11–18 years and found that the presence of NAFLD was associated with higher body weight, WC, HC, ALT, GGTP, HDL, but not with gender, age, glucose concentration, AST, T-CH, LDL-CH, or TG. A negative correlation was detected between irisin and BMI, WC, HC, WHR, ALT and a positive correlation with HDL-CH in the entire study group, including the obese and normal-weight groups. Subgroup analyses revealed that irisin levels were negatively correlated with AST in the control group and positively correlated with BMI and WC in all obese patients without NAFLD and with HDL-CH in obese NAFLD patients [36]. Panagiotou et al. revealed in models adjusted for age, sex and race that irisin was negatively correlated with HDL-CH, but not with BMI, blood pressure, FPG, HbA1c, ALT, AST or physical activity. It should be highlighted that these results apply to patients over 55 years of age with either diabetes or two other cardiovascular risk factors [45]. Considering the correlation between irisin in the whole study population, we found it between BMI, HbA1c, eGFR and HSI (positively) as well as with bilirubin, creatinine, urea and uric acid (negatively). Additionally, the performed statistical analysis showed that BMI, HbA1c and creatinine were the most strongly and significantly correlated with irisin concentration. Recent studies indicate the presence of a strong association between CKD and lowered irisin levels [46,47].

Considering the results we obtained, it is also worth mentioning the previously suggested relationship between a higher concentration of irisin and the progression of NAFLD related to the increase in fat content in the liver, the onset of inflammation and the process of fibrosis. It has been shown that the level of this myokine may indicate a more aggressive liver disease phenotype associated with increased intrahepatic fat content, fibrogenesis and more severe liver damage [39,48]. This may suggest the possibility of including this chemokine in a panel of studies assessing the risk of NASH and its further life-threatening consequences.

As mentioned, some NAFLD patients develop disease progression and development of NASH, liver fibrosis and cirrhosis. It has been shown that polymorphism in the FNDC5 gene (rs3480) is associated with NAFLD progression to NASH and liver fibrosis. It should be highlighted that this effect is independent but additive to the patatin-like phospholipase domain-containing 3 (PNPLA3) and the transmembrane 6 superfamily member 2 (TM6SF2)—two major inherited determinants of hepatic fat accumulation, based on genome-wide association studies [49]. Additionally, PNPLA3 148M allele carriers had higher plasma levels of irisin than the non-carriers [50]. The above data indicate the validity of checking irisin as an NASH biomarker.

The following limitations of our study should also be recognized. Our study is biased by its design, namely a cross-sectional observational study, although the participants were randomly selected. The study included a relatively low number of patients. In our study, we also did not include information about physical activity before inclusion in the study, which might have affected the concentration of irisin. Some studies have suggested that aerobic exercise training promotes irisin secretion [51]. However, a meta-analysis conducted by Fox et al. demonstrated no significant relationship between post-exercise irisin concentration and age, the intensity of aerobic exercise, or the type of exercise training session [52]. However, the participants did not practice physical activity; they were hospitalized for at least two days. The limitation of our study is also the lack of the term FNDC5 polymorphism.

## 5. Conclusions

The obtained results suggest that irisin may be used as a diagnostic biomarker of NAFLD since it correlates with anthropometric and biochemical parameters associated with liver function. The obtained results suggest that irisin may be used as a diagnostic biomarker of NAFLD since it correlates with anthropometric and biochemical parameters associated with liver function.

## Figures and Tables

**Figure 1 biomedicines-10-02253-f001:**
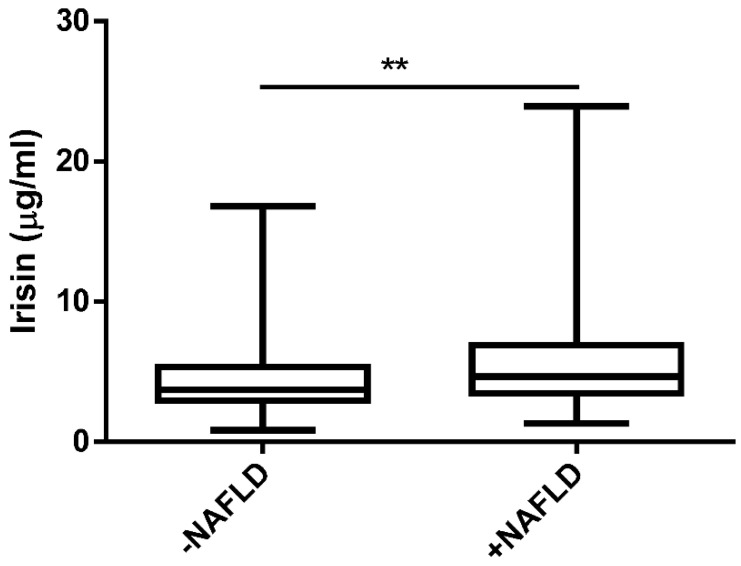
The serum level of irisin in patients without NAFLD (−NAFLD; *n* = 66) and with NAFLD (+NAFLD; *n* = 72), measured by ELISA. The data are expressed as medians with lower and upper quartiles and minimal and maximal values. **, *p* < 0.01 −NAFLD group vs. +NAFLD group.

**Table 1 biomedicines-10-02253-t001:** Anthropometric characteristics of the study groups, including age, sex, blood pressure and presence of T2DM.

Parameter *	Group 0 −NAFLD (*n* = 66)	Group 1 +NAFLD (*n* = 72)	*p* **
Age [years]	67.00 (60.75; 75.00)	65.00 (58.00; 71.75)	0.0632
Body weight [kg]	80.00 (71.50; 90.25)	87.00 (79.00; 99.75)	**0.0007**
Height [cm]	165.00 (160.00; 176.00)	168.00 (160.00; 175.00)	0.6722
WC [cm]	100.50 (94.75; 108.30)	110.00 (101.00; 118.80)	**0.0002**
HC [cm]	106.00 (100.80; 112.00)	112.00 (103.30; 117.80)	**0.0137**
BMI [kg/m^2^]	29.44 (25.57; 31.96)	30.86 (28.72; 34.69)	**0.0015**
WHR	0.95 (0.90; 0.99)	0.98 (0.93; 1.04)	**0.0073**
SBP [mmHg]	130.00 (120.00; 140.00)	135.00 (120.00; 145.00)	0.1712
DBP [mmHg]	80.00 (70.00; 80.00)	80.00 (70.00; 83.75)	0.4875
Sex [% of F]	53	46	0.3983
T2DM [%]	47	64	**0.0456**

* BMI—body mass index, DBP—diastolic blood pressure, F—female, HC—hips circumference, NAFLD—non-alcoholic fatty liver disease, SBP—systolic blood pressure, T2DM—type 2 diabetes mellitus, WC—waist circumference, WHR—waist-hip ratio. ** *p*-value assessed using the Mann–Whitney U test, except for the sex and T2DM variables in which the chi-square test was used. Data is expressed as median (Quartile 1; Quartile 3). The bolded results indicate statistically significant differences.

**Table 2 biomedicines-10-02253-t002:** Biochemical characteristics of the study groups.

Parameter *	Group 0 −NAFLD (*n* = 66)	Group 1 +NAFLD (*n* = 72)	*p* **
FPG [mmol/L]	6.82 (5.30; 10.05)	8.12 (5.64; 10.98)	0.1616
PPG [mmol/L]	8.55 (6.45; 16.45)	9.10 (5.98; 13.21)	0.5080
HbA1c [%]	7.59 (5.70; 9.22)	7.92 (6.10; 9.81)	0.1655
ALT [U/L]	20.50 (14.00; 25.00)	26.00 (18.25; 42.75)	**0.0018**
AST [U/L]	21.00 (17.00; 30.00)	26.50 (19.00; 38.25)	**0.0256**
GGTP [U/L]	25.00 (17.00; 48.78)	48.47 (29.25; 72.25)	**<0.0001**
Total bilirubin [μmol/L]	9.65 (7.40; 16.28)	11.85 (7.83; 19.13)	0.0833
Creatinine [μmol/L]	85.00 (67.75; 110.00)	84.00 (70.25;103.50)	0.9975
Urea [mmol/L]	7.00 (5.20; 9.97)	6.30 (5.25; 8.90)	0.3201
eGFR [ml/min/1.73 m^2^]	72.50 (51.43; 91.75)	72.30 (58.83; 93.00)	0.6168
T-CH [mmol/L]	4.31 (3.80; 5.27)	4.12 (3.51; 5.38)	0.4524
LDL-CH [mmol/L]	2.67 (2.20; 3.26)	2.39 (1.92; 3.20)	0.2136
HDL-CH [mmol/L]	1.04 (0.85; 1.29)	0.92 (0.73; 1.22)	0.0841
TG [mmol/]	1.29 (0.89; 1.78)	1.56 (1.05; 2.31)	**0.0053**
Uric acid [μmol/L]	370.00 (288.80; 436.50)	399.00 (309.00; 470.30)	0.2273
HIS	33.30 (28.10; 37.35)	37.20 (31.88; 41.53)	**0.0007**
FLI	56.69 (35.96; 73.60)	88.75 (58.23; 95.28)	**<0.0001**

* ALT—alanine aminotransferase, AST—aspartic aminotransferase, FPG—fasting plasma glucose, PPG—postprandial plasma glucose, GGTP- gamma-glutamyltransferase, eGFR—estimated glomerular filtration rate, FLI—Fatty Liver Index, HbA1c—glycated hemoglobin, HDL-CH—HDL cholesterol, HSI—Hepatic Steatosis Index, LDL-CH—LDL cholesterol, NAFLD—non-alcoholic fatty liver disease, T-CH—total cholesterol, TG—triglycerides, T2DM—type 2 diabetes mellitus. ** *p*-value assessed using the Mann–Whitney U test. The bolded results indicate statistically significant differences.

**Table 3 biomedicines-10-02253-t003:** Spearman correlations and linear regression analysis of irisin concentration and anthropometric and biochemical parameters in the whole study population (*n* = 138). Only the best irisin-associated variables are shown, as assessed by the stepwise forward multiple regression method.

Parameters *	Rho **	*p* ***	β ± SE ****	*p* ***
Age [years]	−0.105978	0.216040		
Body weight [kg]	0.101933	0.234184		
Height [cm]	−0.162921	0.056230		
Waist [cm]	0.061384	0.474479		
Hip [cm]	0.115337	0.177955		
BMI [kg/m^2^]	**0.187116**	**0.027980**	**0.182 ± 0.084**	**0.031**
WHR	−0.008709	0.919253		
SBP [mmHg]	0.062872	0.463811		
DBP [mmHg]	0.033138	0.699609		
FPG [mmol/L]	0.164347	0.054079		
PPG [mmol/L]	0.128057	0.134441		
HbA1c [%]	**0.274798**	**0.001107**	**0.188 ± 0.085 ^#^**	**0.028**
ALT [U/L]	+0.044641	0.603135		
AST [U/L]	−0.032370	0.706247		
GTP [U/L]	−0.023240	0.786731		
Total bilirubin [μmol/L]	**−0.191931**	**0.024122**		
Creatinine [μmol/L]	**−0.361875**	**0.000013**	**−0.245 ± 0.080 ^#^**	**0.003**
Urea [mmol/L]	**−0.262494**	**0.001869**		
eGFR [mL/min/1.73 m^2^]	**0.303918**	**0.000290**		
TCH [mmol/L]	0.060998	0.477266		
LDL-CH [mmol/L]	−0.005353	0.950311		
HDL-CH [mmol/L]	0.141906	0.096862		
TG [mmol/L]	0.085787	0.317093		
Uric acid [μmol/L]	**−0.232041**	**0.006171**		
HSI	**0.168457**	**0.048262**		
FLI	0.132456	0.121458		

* ALT—alanine aminotransferase, AST—aspartic aminotransferase, BMI—body mass index, DBP—diastolic blood pressure, eGFR—estimated glomerular filtration rate, HDL-CH—HDL cholesterol, HSI—Hepatic Steatosis Index, FLI—Fatty Liver Index, FPG—fasting plasma glucose, GGTP- gamma-glutamyltransferase, HbA1c—glycated hemoglobin, HC—hips circumference, LDL-CH—LDL cholesterol, NAFLD—non-alcoholic fatty liver disease, PPG—postprandial plasma glucose, SBP—systolic blood pressure, T-CH—total cholesterol, TG—triglycerides, WC- waist circumference, WHR—waist-hip ratio. ** Rho—Spearman’s rank correlation coefficient. *** *p*-value. β ± SE ****—regression coefficient ± standard error. ^#^ variables were log-transformed prior to linear regression analysis. The bolded results indicate statistically significant associations.

**Table 4 biomedicines-10-02253-t004:** Results of the general linear model (GLM) for irisin concentration and NAFLD status (with interactions). The models with a sigma-restricted parameterization; effective hypothesis decomposition, ANOVA and ANCOVA test.

	Effect *	SS **	DF ***	MS ****	F *****	*p* ******
Model 1	Intercept	1.34522767	1	1.34522767	33.6206225	**0.000000**
	NAFLD	0.0577242089	1	0.0577242089	1.44267315	0.231874
	Creatinine	0.508549215	1	0.508549215	12.7099238	**0.000508**
	NAFLD * Creatinine	0.240224892	1	0.240224892	6.00382421	**0.015595**
	AST	0.100042825	1	0.100042825	2.5003218	0.116235
	NAFLD * AST	0.246810535	1	0.246810535	6.16841598	**0.014268**
	HbA1c	0.2842315	1	0.2842315	7.10366	**0.008661**
	Error	5.24156934	131	0.0400119797		
Model 2	Intercept	0.557284672	1	0.557284672	12.6422799	**0.000521**
	Urea	0.354331947	1	0.354331947	8.03819643	**0.005290**
	HbA1c	0.222915099	1	0.222915099	5.05693987	**0.026159**
	BMI	0.220387716	1	0.220387716	4.99960493	**0.027006**
	Error	5.90685751	134	0.0440810262		
Model 3	Intercept	0.625281112	1	0.625281112	14.3295681	**0.000230**
	BMI	0.206800496	1	0.206800496	4.73924723	**0.031233**
	Creatinine	0.414002118	1	0.414002118	9.48768707	**0.002510**
	HbA1c	0.21534678	1	0.21534678	4.93510245	**0.027994**
	Error	5.84718734	134	0.0436357264		

* ALT—alanine aminotransferase, AST—aspartic aminotransferase, BMI—body mass index, HbA1c—glycated hemoglobin, NAFLD—non-alcoholic fatty liver disease. ** SS—sums of squares. *** DF—degrees of freedom. **** MS—mean square. ***** F—F test statistic. ****** *p*-value. The bolded results indicate statistically significant associations.

**Table 5 biomedicines-10-02253-t005:** Logistic regression models for NAFLD risk in relation to irisin concentration. Model 1 refers to univariate irisin-NAFLD dependency and model 2 refers to irisin-NAFLD relationship adjusted for variables strongly correlated with irisin concentration.

	Effect *	OR **	*p* ***
Model 1 (unadjusted)	Intercept	0.486 (0.234–1.009)	**0.053**
	Irisin	1.172 (1.028–1.336)	**0.018**
Model 2 (adjusted)	Intercept	0.014 (0.001–0.214)	**0.002**
	Irisin	1.143 (0.994–1.314)	**0.061**
	Creatinine	1.001 (0.994–1.009)	0.734
	HbA1c	0.967 (0.810–1.155)	0.711
	BMI	1.134 (1.045–1.231)	**0.003**

* BMI—body mass index, HbA1c—glycated hemoglobin. ** OR—odd ratios. *** *p*-value. The bolded results indicate statistically significant associations.

## Data Availability

Marcin Kosmalski will provide data.

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
