# Peer review of "Irisin Is Related to Non-Alcoholic Fatty Liver Disease (NAFLD)"

_biomedicines, 2022, doi:10.3390/biomedicines10092253_

Round 1

Reviewer 1 Report

Marcin et al. shows that serum irisin is elevated in NAFLD patients and the myokine may be related to the pathogenesis of NAFLD.  This study is novel and significant but this manuscript needs minor improvement.

1. Irisin is significantly higher in NAFLD patients but is not correlated to HSI, FLI, and transaminase.  It is unclear how irisin is useful in the pathogenesis of NAFLD; please give your thoughts on whether it is useful in the diagnosis of NAFLD, in predicting prognosis, or if it has other usefulness.

2.   Multivariate analysis should also be performed to examine variables associated with Irisin.

3.  The line breaks are disrupting the style of Table1 and Table2. Please correct.

Author Response

Dear Professor,

We would like to thank the Reviewer for insightful comments on our manuscript. We appreciate the time and effort that was dedicated to provide the feedback on our manuscript, and we are grateful for valuable suggestions leading to vast improvement of our manuscript. We have now responded point-by-point to the comments and concerns that were raised. We feel that the revised manuscript is clearer, more accurately reflects the aim of review, provides more in-depth analysis of presented facts with their critical discussion and interpretations. Final conclusions with future recommendations for human have been added.

,, Marcin et al. shows that serum irisin is elevated in NAFLD patients and the myokine may be related to the pathogenesis of NAFLD.  This study is novel and significant but this manuscript needs minor improvement.’’ 

Comment (C): ,, Irisin is significantly higher in NAFLD patients but is not correlated to HSI, FLI, and transaminase.  It is unclear how irisin is useful in the pathogenesis of NAFLD; please give your thoughts on whether it is useful in the diagnosis of NAFLD, in predicting prognosis, or if it has other usefulness.’’

Answer (A): We would like to thank the Reviewer for his valuable remark regarding the need to perform variables associated with irisin. We observed the association between the concentration of irisin and HSI, FLI and ALT, which we did not observe in linear regression. We are aware that previous studies on the level of irisin in NAFLD patients are divergent, but indicate the involvement of this myokine in etiopathogenesis. As mentioned above, these differences may be related to different NAFLD diagnosis tools, different patient groups and irisin level tests. In addition, it should be taken into account that liver disease does not progress in all patients. The time of their appearance is also unknown, which may also affect the level of this myokine. The differences between linear and multiple regression regarding the level of irisin and HSI and FLI require further studies (for example, in a larger group of patients or the need to consider NAFLD diagnostic tools other than HSI, FLI and US) to evaluate irisin as a diagnostic tool for large-scale use in everyday clinical practice.

Comment (C): ,,Multivariate analysis should also be performed to examine variables associated with Irisin.’’

Answer (A):  Thank you for your valuable remark. As you indicated we carried out multivariate analysis (MANOVA) and added short description of obtained results. ,, Table 5. presents result of multiple ANOVA (MANOVA). We observed the association between the concentration of irisin and age, height, body weight, BMI, WC, HC, WHR, SBP, DBP, eGFR, FPG, PPG, HbA1c, uracic acid, TCH, LDL-CH, HDL-CH, TG, ALT, HIS and  FLI. There was no significant association between irisin con-centration and urea, creatinine, total bilirubin, AST, GGTP.’’

Comment (C): ,, The line breaks are disrupting the style of Table1 and Table2. Please correct..’’

Answer (A): Your remark is pertinent. We corrected the style of Table 1 and Table 2.

Reviewer 2 Report

The paper “Irisin is related with non-alcoholic fatty liver disease (NAFLD)”, proposes an interesting investigation on the possible role of irisin in the development of NAFLD, with also a potential value as supportive prognostic/diagnostic marker. The paper is well written and the study on the patients performed in a very accurate way, in particular as to the selection of suitable candidates. These are however same points, mainly as to data presentation in the Results, which need to be better clarified, as detailed below. For this reason, a minor revision is required to improve the paper.

General remarks

Page 2, the sentence beginning as: “Irisin is a myokine release….” Should be exchanged with the following one “This hormone is involved in metabolic pathway…”, in order to state first that irisin is a hormone and provide its description, followed by its roles in normal and altered physiological states of organs. A short statement of what is irsin should be also added at the beginning of the Abstract

In all tables 1—4 it is to explain the reason of red characters. It seems that it is indicating a significant difference for the related data in the same line. If they are used with a somewhat meaning, this should be clearly explained at the footnote of each table, although it is to verify is this kind of “labelling” is admitted by the Journal style. Also, check the use of asteriscs, between abbreviations at the heading of the tables and their recall in the related footnotes.

Tables 1,2 - data in parenthesis seem to be the span of data reported as “median?” Please clarify in the table heading sentence. Also, check with the Journal style, maybe they are reported with a line instead of “;”

Table 3- the meaning of RHO is missing at the footnote of the table.

Tables 3 and 4 - please provide for each table its relevant information, similarly to Tables 1 and 2.

Minor remark- please correct y axis label in the Figure 1. "micron" seems in bold

Author Response

Dear Professor,

We would like to thank the Reviewer for insightful comments on our manuscript. We appreciate the time and effort that was dedicated to provide the feedback on our manuscript, and we are grateful for valuable suggestions leading to vast improvement of our manuscript. We have now responded point-by-point to the comments and concerns that were raised. We feel that the revised manuscript is clearer, more accurately reflects the aim of review, provides more in-depth analysis of presented facts with their critical discussion and interpretations. Final conclusions with future recommendations for human have been added.

,, The paper “Irisin is related with non-alcoholic fatty liver disease (NAFLD)”, proposes an interesting investigation on the possible role of irisin in the development of NAFLD, with also a potential value as supportive prognostic/diagnostic marker. The paper is well written and the study on the patients performed in a very accurate way, in particular as to the selection of suitable candidates. These are however same points, mainly as to data presentation in the Results, which need to be better clarified, as detailed below. For this reason, a minor revision is required to improve the paper.

General remarks: ’’

Comment (C):, Page 2, the sentence beginning as: “Irisin is a myokine release….” Should be exchanged with the following one “This hormone is involved in metabolic pathway…”, in order to state first that irisin is a hormone and provide its description, followed by its roles in normal and altered physiological states of organs. A short statement of what is irsin should be also added at the beginning of the Abstract’’

Answer (A): Thank you for your valuable remark. We exchanged the sentence in page 2 and we added short statement at the beginning of Abstract.  

,, Irisin is a hormone involved in metabolic pathways, namely it increases the uptake of glucose and fatty acids by muscles, reduces gluconeogenesis and stimulates glycogenesis in the liver, as well as converts white adipose tissue (WAT) into brown adipose tis-sue. Irisin was also demonstrated to diminish the severity of inflammation as well as to affect the function of kidneys, neurons, bones, endothelial cells and beta cells of the pancreas [15,16].’’

,,Irisin is a cytokine involved in many metabolic pathways occuring among others in muscles, adipose tissue and liver. Thus fluctuations of irisin level are suggested to be related to some diseases.’’

Comment (C): ,, In all tables 1—4 it is to explain the reason of red characters. It seems that it is indicating a significant difference for the related data in the same line. If they are used with a somewhat meaning, this should be clearly explained at the footnote of each table, although it is to verify is this kind of “labelling” is admitted by the Journal style. Also, check the use of asteriscs, between abbreviations at the heading of the tables and their recall in the related footnotes.’’

Answer (A): We removed red characters and checked the Journal style.

Comment (C): ,,Tables 1,2 - data in parenthesis seem to be the span of data reported as “median?” Please clarify in the table heading sentence. Also, check with the Journal style, maybe they are reported with a line instead of “;”’’

Answer (A): Thank you for your remark. We added how data is expressed. We checked Journal style.

,, Data is expressed as median (Quartile 1; Quartile 3)’’

Comment (C): ,, Table 3- the meaning of RHO is missing at the footnote of the table.’’

Answer (A): Thank you for your apt remark. We added the meaning of RHO.

,, *** Rho – Spearman’s rank correlation coefficient’’

Comment (C): ,, Tables 3 and 4 - please provide for each table its relevant information, similarly to Tables 1 and 2.’’

Answer (A): Thank you for your valuable remark. We added relevant information to each table similarly to Table 1 and 2.

Comment (C): ,, Minor remark- please correct y axis label in the Figure 1. "micron" seems in bold’’

Answer (A): Thank you for your remark. We checked the style of y axis. The style of entire y axis is bold.